# Gait Adaptation Is Different between the Affected and Unaffected Legs in Children with Spastic Hemiplegic Cerebral Palsy While Walking on a Changing Slope

**DOI:** 10.3390/children9050593

**Published:** 2022-04-22

**Authors:** Tae Young Choi, Dongho Park, Dain Shim, Joong-on Choi, Juntaek Hong, Yongjin Ahn, Eun Sook Park, Dong-wook Rha

**Affiliations:** 1Department of Rehabilitation Medicine, Research Institute of Rehabilitation Medicine, Severance Hospital, Yonsei University College of Medicine, Seoul 03722, Korea; piyo1214@yuhs.ac (T.Y.C.); dpark@gatech.edu (D.P.); sdi3807@yuhs.ac (D.S.); supermanjo12@yuhs.ac (J.-o.C.); ghdwnsxor@yuhs.ac (J.H.); ahnmedic1@yuhs.ac (Y.A.); pes1234@yuhs.ac (E.S.P.); 2Woodruff School of Mechanical Engineering, Georgia Institute of Technology, Atlanta, GA 30332, USA; 3Institute of Robotics and Intelligent Machines, Georgia Institute of Technology, Atlanta, GA 30332, USA

**Keywords:** cerebral palsy, gait analysis, slope walking, spastic hemiplegia, kinematics, biomechanics

## Abstract

Walking on sloped surfaces requires additional effort; how individuals with spastic hemiplegic cerebral palsy (CP) manage their gait on slopes remains unknown. Herein, we analyzed the difference in gait adaptation between the affected and unaffected legs according to changes in the incline by measuring spatiotemporal and kinematic data in children with spastic hemiplegic CP. Seventeen children underwent instrumented three-dimensional gait analysis on a dynamic pitch treadmill at an incline of +10° to −10° (intervals of 5°). While the step length of the affected legs increased during uphill gait and decreased during downhill gait, the unaffected legs showed no significance. During uphill gait, the hip, knee, and ankle joints of the affected and unaffected legs showed increased flexion, while the unaffected leg showed increased knee flexion throughout most of the stance phase compared with the affected leg. During downhill gait, hip and knee flexion increased in the affected leg, and knee flexion increased in the unaffected leg during the early swing phase. However, the ankle plantar flexion increased during the stance phase only in the unaffected leg. Although alterations in temporospatial variables and joint kinematics occurred in both legs as the slope angle changed, they showed different adaptation mechanisms.

## 1. Introduction

Children with spastic cerebral palsy (CP) show different abnormal gait patterns due to varying degrees of muscle weakness, spasticity, and motor incoordination [1]. Changes in the ground environment, such as uphill and downhill slopes, require additional efforts and different compensation strategies during ambulation.

There have been several previous studies on the kinetic and kinematic changes accompanying various ground environments. Adults without neurological impairments can negotiate ramps by adapting differently between uphill and downhill slopes [2,3,4,5,6,7]. Previous studies comparing children with spastic diplegic CP and typically developing (TD) peers during uneven-level gait have shown similar adaptation mechanisms between the two groups; however, children with CP required additional efforts. During uphill gait, children with spastic diplegic CP showed increased trunk forward-leaning [8,9], knee flexion at initial contact [8,9,10,11], and increased hip range of motion in the sagittal plane [11] compared with TD peers. Kinematic analysis of the downhill gait of children with spastic diplegic CP showed pronounced backward trunk-leaning and increased hip extension at initial contact [9] than TD peers. Ankle joints showed controversial results, revealing more dorsiflexion [9] or plantar flexion [11,12] at initial contact during downhill gait.

Gait analysis of children with spastic hemiplegic CP on leveled ground showed slower walking velocity [13], shorter stride length [14], and greater hip and knee flexion of the affected and unaffected legs at initial contact and the swing phase [13] compared with those of TD peers, lowering the center of mass level to compensate for stability. When comparing limbs, the unaffected leg had a longer stance time and stance phase percentage [13,15] than the affected leg.

Although there have been studies on ambulation at different slopes in healthy adults and patients with spastic diplegic CP, and even level gait in spastic hemiplegic CP as mentioned above, little is known about how children with hemiplegic CP manage their gait on slopes, especially when comparing adaptation mechanisms between the affected and unaffected limbs. We hypothesized that the affected and unaffected legs would have similar adaptation mechanisms in spastic diplegic CP and TD peers; however, the affected and unaffected legs would show different adaptation mechanisms as the slope angle steepens.

## 2. Materials and Methods

### 2.1. Study Design

This prospective study was conducted at a university-affiliated hospital. The Institutional Review Board of Yonsei University Health System, Severance Hospital, granted ethical approval for this study (4-2020-1182). All parents and children were informed of the purpose and protocol of the study before enrollment and written informed consent was obtained from all participants and/or their parents. The trial was registered with the Clinical Research Information Service (Identifier No. KCT0006927).

### 2.2. Participants

Children with spastic hemiplegic CP who met the inclusion criteria were recruited for this study. The inclusion criteria were as follows: (1) age between 6 and 18 years, (2) ability to walk independently on a leveled surface and slope, and (3) ability to obey commands and perform tasks. The exclusion criteria were as follows: (1) children who underwent orthopedic surgery within 1 year, (2) children who were administered botulinum toxin injection within 6 months, and (3) children who needed walking aids.

A sample size calculation was performed using a two-tailed paired t-test based on a previous study dealing with the kinematic difference between spastic diplegic CP and TD on inclined surfaces [10]. This indicated that a sample size of 18 was sufficient to detect a difference of 10° of knee flexion angle at initial contact on a 10° uphill slope, assuming a standard deviation of 12°, a power of 90%, and a significance level of 5%. This number was increased to 20 to allow a predicted dropout rate of approximately 10%.

### 2.3. Gait Protocol

The participants walked at a self-selected comfortable speed on a dynamic pitch treadmill embedded with a force plate to measure the ground reaction force (M-gait, Motek, The Netherlands) (Figure 1). A single stride was recorded when a moment greater than 200 Nm/kg was measured consecutively by both feet using the D-flow software (Motek, The Netherlands). Upon 20 successful strides, the software enabled slope changes. During the uphill trial, the slope angle was increased by 5° degrees from 0° to 10° and during the downhill trial, the slope angle was decreased by 5° degrees from 0° to −10°.

### 2.4. Gait Analysis

Gait analysis was performed using a computerized three-dimensional (3D) motion capture system with 10 infrared cameras (Vicon Motion System, Oxford, UK) at a sampling rate of 100 Hz. For each participant, 26 reflective markers were attached to their lower limbs according to the Human Body Model [16]. Five middle strides were used for the analysis at each slope angle. Gait offline analysis tool software (Motek, Amsterdam, Netherlands) was used to compute temporospatial parameters and joint kinematics of the hip, knee, and ankle in the sagittal plane.

### 2.5. Statistical Analysis

A repeated measures analysis of variance (ANOVA) (slope angle × limb) with Bonferroni adjustment for multiple comparisons was performed to compare temporospatial variables of the three slope angles on uphill and downhill gaits.

We performed a one-way repeated measures ANOVA for each leg to compare the joint kinematics of different slope angles. We also performed a paired *t*-test as a post hoc analysis to compare two different slope angles (0° vs. 5°, 5° vs. 10°, and 0° vs. 10°, α = 0.05). For each ANOVA and paired t-test, we created a statistical parametric mapping (SPM){F} curve from the conventional univariate F-statistic and an SPM{t} curve from the conventional univariate t-statistic at each time point in the gait cycle [17]. We computed the post hoc SPM{t} to compare two different slope angles when the SPM{F} curve exceeded the set threshold. We created suprathreshold clusters in sections of the cycle to show significant differences between the two different slope angles when the SPM{t} curve exceeded the set threshold in the gait cycle. We used the Bonferroni correction to adjust α for multiple comparisons. The level of significance was set at *p* < 0.05. All analyses were performed in Matlab2019 (The Mathworks, Inc., Natick, MA, USA) using SPM1d (vM.0.4.8). All computational details are available in the open-source software at “SPM1d. Available online: https://spm1d.org (accessed on 18 September 2021)”.

## 3. Results

Twenty children with spastic hemiplegic CP were recruited for this study. As three children could not complete the experimental protocol, 17 children were finally included in this study. The general characteristics of the participants are presented in Table 1. One participant used an orthosis intermittently for daily activities, and one participant underwent orthopedic surgery 3 years ago. Twelve children had a history of botulinum toxin injection more than 6 months before the study, and the injection administration ranged from 1 to 11 times.

### 3.1. Gait Adaptation during Uphill Gait

The affected leg showed longer step lengths in terms of the interaction effect of the limb compared with the unaffected leg (*p* < 0.01). The unaffected leg did not show any statistical significance in the step length during a change in the inclination angle (Table 2).

For kinematic variables, the affected and unaffected legs showed an overall increasing tendency for flexion of the hip, knee, and ankle joints. While the hip and ankle joints of both limbs showed similar results, with increased flexion during most of the gait cycle except the pre-swing phase as the slope angle steepened (*p* < 0.05), the knee joint showed a significant difference between the affected and unaffected legs. Only the unaffected legs showed more flexion during the midstance to terminal stance phase (*p* < 0.05) (Figure 2).

### 3.2. Gait Adaptation during Downhill Gait

While the stance time of both limbs showed a significant decrease (*p* < 0.01), the swing time of both limbs did not show a significant change as the slope angle increased. The affected leg showed shorter step lengths as the slope deepened (*p* < 0.01), whereas the unaffected leg did not show a significant change (Table 3).

Analysis of kinematic variables showed more flexion of the affected leg at the hip and knee joints during the early swing phase, whereas the unaffected leg only showed increased flexion of the knee joints (*p* < 0.05). The ankle plantar flexion increased in the affected and unaffected legs during the terminal swing phase (*p* < 0.05). However, only the unaffected ankle showed greater plantar flexion during the stance phase (*p* < 0.05) (Figure 3).

## 4. Discussion

Walking on a sloped surface requires additional effort, and gait adjustments can be made by modifying neural control, thereby inducing different biomechanical responses [2,3,4,5,6,7]. Children with spastic diplegic CP could negotiate ramps by adapting similarly to adults without pathology or TD peers; however, this requires additional adaptation mechanisms [8,9,10,11,12]. Children with spastic hemiplegic CP also need extra effort to adapt their gait pattern to walk up an inclined surface. The inclined gait pattern could be different between the legs because affected legs have various deficits in neural coordination [1], muscle tone [18], and strength [2,4,18].

We performed 3D computer gait analysis in uphill and downhill gait conditions, using a dynamic pitch treadmill to investigate the adaptation mechanism of the affected and unaffected limbs of children with spastic hemiplegic CP according to the change in the slope angle during gait. The current study showed a significant difference in the adaptation mechanism between the affected and unaffected limbs in the temporospatial parameters and joint kinematics of the hip, knee, and ankle in the sagittal plane.

The step length of the affected leg in spastic hemiplegic CP showed adaptation to the change in the inclination angle that increased during uphill gait in terms of the interaction effect of the limb and decreased during downhill gait. However, the unaffected leg showed no statistical significance. Previous studies analyzing the relationship between energy generation and step length in hemiplegic stroke patients showed that stance leg forward propulsion is an important factor in generating step length [19,20]. In addition, patients with longer step length in the affected leg than the unaffected leg had a less affected leg plantar flexor moment with increased unaffected leg ankle impulse [21,22]. Another study on spastic hemiplegic CP reported that children with CP did not perform sufficient work when the affected leg trailed during double support; therefore, additional positive work was needed in the unaffected leg during single support [23]. As upslope walking requires an additional propulsion force to propel the body upward, the unaffected leg requires additional work to compensate for the impaired propulsion of the affected leg during uphill gait. This trend might explain the reason for generating longer step length in the affected limb of children with spastic hemiplegic CP in the present study.

Downslope walking can be viewed from a different perspective. Increased dynamic instability and a higher probability of slip-related falls are important factors in downslope walking. Previous studies analyzing temporospatial variables and gait stability have shown that a shorter stride length [24,25,26,27,28,29] and increased stride frequency [24,28,29] could place the center of mass closer to the leading foot, thereby increasing dynamic stability. Better stability could be obtained by decreasing the step length of the affected limb, resulting in decreased step length during downhill gait as the slope angle steepened in our study.

Kinematic analysis during uphill gait agreed with previous studies of healthy adults [3,5,6,7], children with spastic diplegic CP, and children with TD [8,9,10,11,12]. Previous studies showed increased hip flexion, knee flexion, and ankle dorsiflexion during the gait cycle, thereby raising the limb for toe clearance and heel strike. Moreover, the lower limbs progressively extended during midstance to propel the body up the incline. However, the unaffected leg showed increased knee flexion during the longer portion of the stance phase compared with the affected leg. The difference between the affected and unaffected legs may indicate that better control of the unaffected leg was used to adapt to the slope angle change.

A larger significant portion during the gait cycle was also observed when analyzing ankle joints in the sagittal plane during downhill gait, as the unaffected leg showed an increase in ankle plantar flexion during the stance phase compared with the affected leg, which could be explained by the same reason as mentioned above. Moreover, the ankle kinematics of the affected legs were variable compared with those of the unaffected legs in this study (Figure 3). Since we recruited children with diverse ankle joint motions during gait, it would have been challenging to produce consistent changes. Lastly, as previous studies of children with spastic diplegic CP and TD showed controversial results for ankle joints on descent [9,11,12], the mechanism is poorly understood, and further investigation is needed.

Analysis of the downhill gait also showed different adaptation mechanisms at the hip and knee joints in the sagittal plane. While the affected leg showed increased hip and knee flexion during the early swing phase, the unaffected leg only showed increased knee flexion during the same phase. Increased knee flexion in both limbs during the pre-swing phase has been shown to lower the foot contact on descent for better adjustment in healthy adults [2,4] and spastic diplegic CP peers [9], and a similar adaptation mechanism would have occurred in spastic hemiplegic CP. However, increased hip joint flexion of the affected leg during the pre-swing phase was not observed in previous studies of adults with spastic diplegic CP and healthy adults during downhill gait. Impairments in ankle plantar flexion during the pre-swing phase can lead to inappropriate swing phase initiation, and the hip flexors are additionally used to compensate for hemiplegia [22,30,31]. As the leg affected with spastic hemiplegic CP did not perform sufficient push-off, extra hip flexion may be needed to compensate for insufficient swing initiation. Therefore, electromyography measurements might help explain the underlying mechanism in future studies.

However, this study has some limitations. First, joint kinematics in the coronal and transverse planes and joint kinetics were not analyzed in our study. If further studies measure these additional joint kinematics and kinetics, it could help understand the biomechanical strategy to adapt to the slope. In addition, electromyography measurements of leg muscles might explain muscle activation patterns during inclined gait more comprehensively. Second, patients with spastic hemiplegic CP with various gait patterns were recruited in our study. Previous studies subdivided gait patterns of spastic hemiplegia into four groups based on the sagittal plane kinematics and electromyographic data [32,33]; Type 1 showed “drop foot” during the swing phase due to inappropriate control of the ankle dorsiflexors. Type 2 showed ankle plantar flexion throughout the gait cycle due to triceps surae contracture, spasticity, and impaired ankle dorsiflexion. Type 3 showed flexed, “jump knee gait” as a result of hamstring and rectus femoris muscle co-contraction, and Type 4 showed proximal involvement of the pelvic and hip joints. Patients with different gait patterns may also show different adaptation patterns in both legs during a slope gait. Further study is required to compare which adaptation patterns are shown regarding their underlying gait patterns.

To the best of our knowledge, this is the first study to analyze the differences in gait adaptation patterns between the legs of children with spastic hemiplegic CP according to changes in the slope angle, which was different from slope gait adaptation mechanism of spastic diplegic CP and TD peers. Consideration of this unique adaptation pattern would be helpful to provide a specific intervention program for hemiplegic CP. For example, training focused on promoting propulsion strength of the affected leg might reduce the asymmetry of gait during incline walking in children with hemiplegic CP. However, further clinical study is needed to ascertain whether these interventions can improve gait patterns on slopes.

## 5. Conclusions

The results of this study demonstrated that children with spastic hemiplegic CP negotiated inclined surfaces by adapting their affected and unaffected legs differently. Further well-established studies with larger populations for each gait pattern classification are needed to better elucidate the mechanism during uneven-level gait and inspiration for appropriate intervention.

## Figures and Tables

**Figure 1 children-09-00593-f001:**
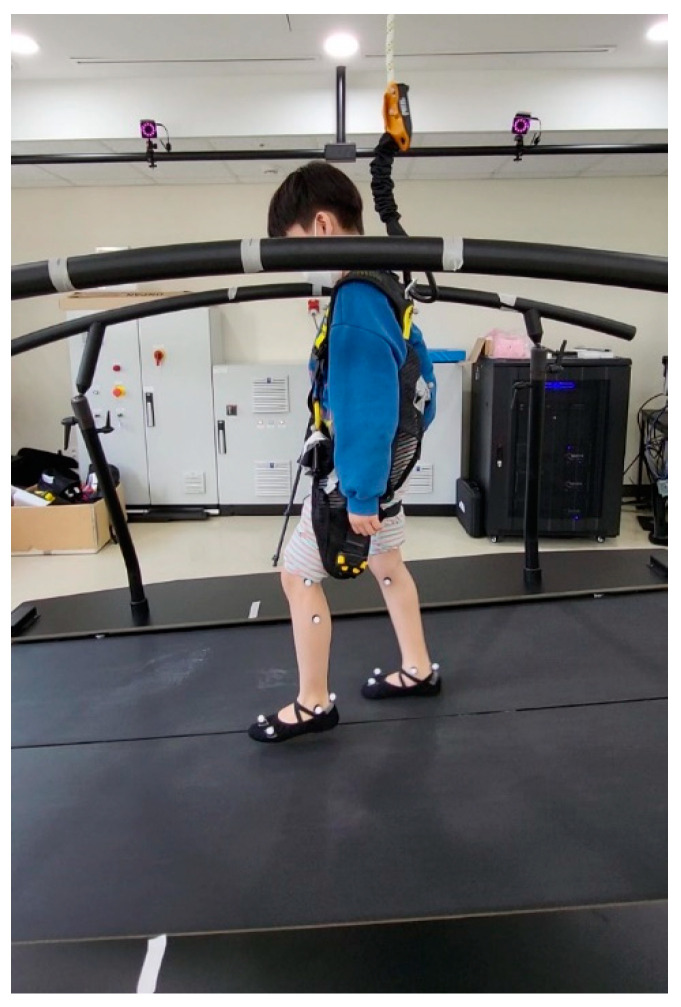
Gait analysis on a dynamic pitch treadmill using a computerized three-dimensional motion capture system. Twenty-six reflective markers were attached to the lower limbs according to the Human Body Model. The participants walked 20 strides for each ramp on both uphill and downhill trials. They were observed prior to inclination change and wore the harness for safety during the study.

**Figure 2 children-09-00593-f002:**
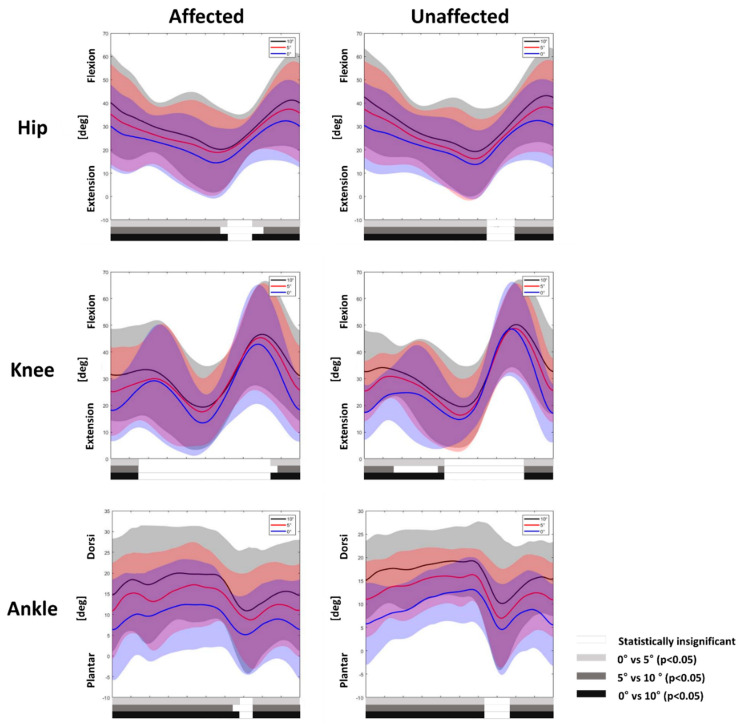
Joint kinematics of the hip, knee, and ankle joints of the affected and unaffected legs during uphill gait. Mean kinetic angles were plotted for 0° (blue), 5° (red), and 10° (black), and the shaded areas indicate ±1 standard deviation. Statistically significant differences are represented by bars below the curves analyzed by statistical parametric mapping; 0° vs. 5° (light gray bars), 5° vs. 10° (gray bars), and 0° vs. 10° (black bars). Statistically significant differences between incline gaits are represented by the white bars.

**Figure 3 children-09-00593-f003:**
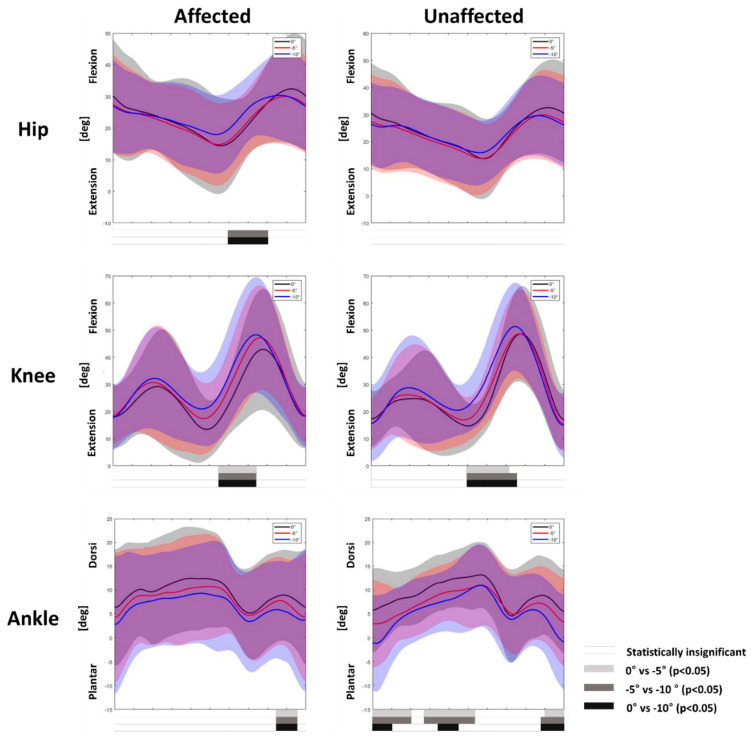
Joint kinematics of hip, knee, and ankle joints of the affected and unaffected legs during uphill gait. Mean kinetic angles were plotted for 0° (black), −5° (red), and −10° (blue), and the shaded areas indicate ±1 standard deviation. Statistically significant differences are represented by bars below the curves analyzed by statistical parametric mapping; 0° vs. −5° (light gray bars), −5° vs. −10° (gray bars), and 0° vs. −10° (black bars). Statistically significant differences between incline gaits are represented by the white bars.

**Table 1 children-09-00593-t001:** Participant characteristics.

Characteristics	Number/Value *
Number of participants	17
Sex, male:female	12:5
Age at assessment (years)	10.0 ± 1.8 (7–13)
Involved side, right:left	7:10
Affected plantarflexor tone (MAS, G1:G1+:G2)	11:3:3
Affected dorsiflexor range of motion	15.9 ± 6.9 (0–20)
GMFCS, I:II	15:2

* Values are expressed as the mean ± standard deviation (range) or number of participants; MAS, Modified Ashworth Scale; GMFCS, Gross Motor Functional Classification System.

**Table 2 children-09-00593-t002:** Results of temporospatial parameters for both limbs during uphill gait.

Slope Angle		0°	5°	10°	*p*-Value	
					Slope Angle	Slope Angle × Limb
Step length (m)	Affected leg	0.280 (0.024)	0.306 ^a^ (0.022)	0.314 ^a^ (0.028)	<0.01 ^b^	0.01 ^b^
	Unaffected leg	0.294 (0.025)	0.289 (0.027)	0.292 (0.029)	0.74	
Swing time (s)	Affected leg	0.363 (0.012)	0.376 (0.013)	0.387 (0.020)	0.16	0.69
	Unaffected leg	0.352 (0.013)	0.355 (0.014)	0.370 (0.015)	0.19	
Stance time (s)	Affected leg	0.821 (0.039)	0.845 (0.043)	0.884 (0.041)	0.11	0.96
	Unaffected leg	0.839 (0.039)	0.861 (0.039)	0.898 (0.041)	0.08	
Stance phase (%)	Affected leg	69.048 (0.689)	68.805 (0.938)	69.349 (1.057)	0.82	0.81
	Unaffected leg	70.176 (0.692)	70.553 (0.679)	70.633 (0.538)	0.57	
Swing phase (%)	Affected leg	30.952 (0.689)	31.195 (0.938)	30.651 (1.057)	0.82	0.81
	Unaffected leg	29.824 (0.692)	29.447 (0.679)	29.367 (0.538)	0.57	

Data are presented as the estimated marginal mean (standard error). Gait conditions: positive values imply an uphill gait, and 0° indicates gait on even ground. ^a^ *p* < 0.05, as determined by Bonferroni-adjusted post hoc analysis compared with 0°. ^b^ *p* < 0.05, as determined by repeated measures analysis of variance.

**Table 3 children-09-00593-t003:** Results of temporospatial parameters for both limbs during downhill gait.

Slope Angle		0°	−5°	−10°	*p*-Value	
					Slope Angle	Slope Angle × Limb
Step length (m)	Affected leg	0.280 (0.024)	0.257 (0.023)	0.245 ^a^ (0.023)	<0.01 ^b^	0.46
	Unaffected leg	0.294 (0.025)	0.280 (0.029)	0.278 (0.028)	0.13	
Swing time (s)	Affected leg	0.363 (0.012)	0.355 (0.013)	0.349 (0.013)	0.29	0.52
	Unaffected leg	0.352 (0.015)	0.331 (0.015)	0.333 (0.015)	0.10	
Stance time (s)	Affected leg	0.821 (0.039)	0.757 ^a^ (0.037)	0.744 ^a^ (0.033)	<0.01 ^b^	0.13
	Unaffected leg	0.839 (0.039)	0.796 (0.035)	0.741 ^a,c^ (0.036)	<0.01 ^b^	
Stance phase (%)	Affected leg	69.048 (0.689)	67.758 (0.763)	67.827 (0.719)	0.10	0.17
	Unaffected leg	70.176 (0.692)	70.582 (0.925)	68.801 (0.812)	0.04 ^b^	
Swing phase (%)	Affected leg	30.952 (0.689)	32.242 (0.763)	32.173 (0.719)	0.10	0.17
	Unaffected leg	29.824 (0.692)	29.418 (0.925)	31.199 (0.812)	0.04 ^b^	

Data are presented as the estimated marginal mean (standard error). Gait conditions: negative values imply downhill gait, and 0° indicates gait on an even ground. ^a^ *p* < 0.05, as determined by Bonferroni-adjusted post hoc analysis compared with 0°. ^b^ *p* < 0.05, as determined by repeated measures analysis of variance. ^c^ *p* < 0.05, as determined by Bonferroni-adjusted post hoc analysis compared with −5°.

## Data Availability

The data presented in this study are available on request from the corresponding author. The data are not publicly available due to ethical reasons.

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
