# Peer review of "Gait Adaptation Is Different between the Affected and Unaffected Legs in Children with Spastic Hemiplegic Cerebral Palsy While Walking on a Changing Slope"

_children, 2022, doi:10.3390/children9050593_

Round 1

Reviewer 1 Report

The authors research the differences in gait adaptation between the affected leg and the unaffected leg of hemiplegic children in different degrees of uphill or downhill.

The authors should describe in a little more detail the methods (and epuipment) used in the research. A better development of the conclusions and repercussions of the research is also needed.

Author Response

Dear reviewer

We sincerely appreciate your comments that proved valuable in improving the quality of our paper. We have addressed all the comments and revised our manuscript accordingly. Our responses to all the comments and the major revisions are listed below.

Point 1: Reply to Response 1. The authors should describe in a little more detail the methods (and epuipment) used in the research.

Response 1: Thank you for this comment. As you pointed out, the gait protocol and the gait analysis that I described in the materials and methods section did not give a detailed explanation. I have modified the contents as follows.

Before (Line in 86-90 before modification)

The participants walked at a self-selected comfortable speed on a dynamic pitch treadmill (M-gait, Motek, Netherlands) (Figure 1). The experiment comprised an uphill trial that went up from 0° to 10° and a downhill trial that went down from 0° to -10° at intervals of 5°, and the slope angle changed when the participants walked 20 steps for each ramp in both experiments.

After (Line in 84-90 after modification)

The participants walked at a self-selected comfortable speed on a dynamic pitch treadmill embedded with a force plate to measure the ground reaction force (M-gait, Motek, Netherlands) (Figure 1). A single stride was recorded when a moment greater than 200Nm/kg was measured consecutively by both feet using the D-flow software (Motek, Netherlands). Upon 20 successful strides, the software enabled slope changes. During the uphill trial, the slope angle was increased by 5° degrees from 0° to 10° and during the downhill trial, the slope angle was decreased by 5° degrees from 0° to -10°.

Before (Line in 93-95 before modification)

Figure 1. Gait analysis on a dynamic pitch treadmill using a computerized three-dimensional motion analysis system. Twenty-six reflective markers were attached to the lower limbs according to the Human Body Model. The participants walked 20 steps for each ramp on both uphill and downhill trials.

After (Line in 96-98 after modification)

Figure 1. Gait analysis on a dynamic pitch treadmill using a computerized three-dimensional motion capture system. Twenty-six reflective markers were attached to the lower limbs according to the Human Body Model. The participants walked 20 strides for each ramp on both uphill and downhill trials.

Before (Line in 98-102 before modification)

Gait analysis was performed using a computerized three-dimensional (3D) motion analysis system (Vicon Vero; Oxford Metrics Inc., Oxford, UK) at a sampling rate of 100 Hz. For each participant, 26 reflective markers were attached to their lower limbs according to the Human Body Model [16]. Five middle steps were used for the analysis at each slope angle.

After (Line in 101-105 after modification)

Gait analysis was performed using a computerized three-dimensional (3D) motion capture system with 10 infrared cameras (Vicon Motion System, Oxford, UK) at a sampling rate of 100 Hz. For each participant, 26 reflective markers were attached to their lower limbs according to the Human Body Model [16]. Five middle strides were used for the analysis at each slope angle.

Point 2: Reply to Response 1. A better development of the conclusions and repercussions of the research is also needed.

Response 1: Thank you for this comment. The manuscript before revision did not explain the influence and contribution to the spastic hemiplegic cerebral palsy population. To give a better explanation of patient care, I added the paragraph below.

Before (Line 264-266 before modification)

To the best of our knowledge, this study is the first to analyze the differences in gait adaptation patterns between the legs of children with spastic hemiplegic CP according to changes in the slope angle.

After (Line in 263-265 after modification) : Deleted

After (Line in 281-289 after modification)

To the best of our knowledge, this is the first study to analyze the differences in gait adaptation patterns between the legs of children with spastic hemiplegic CP according to changes in the slope angle, which was different from slope gait adaptation mechanism of spastic diplegic CP and TD peers. Consideration of this unique adaptation pattern would be helpful to provide a specific intervention program for hemiplegic CP. For example, training focused on promoting propulsion strength of the affected leg might reduce the asymmetry of gait during incline walking in children with hemiplegic CP. However, further clinical study is needed whether these interventions can improve gait patterns on slopes.

Before (Line 284-286 before modification)

Further well-established studies with larger populations for each gait pattern classification are needed to better elucidate the mechanism during uneven-level gait.

After (Line in 293-296 after modification)

Further well-established studies with larger populations for each gait pattern classification are needed to better elucidate the mechanism during uneven-level gait and inspiration for appropriate intervention.

Reviewer 2 Report

Thank you for allowing me to review it. Consider explaining what this study would contribute to patient care.

Author Response

Dear reviewer

We sincerely appreciate your comments that proved valuable in improving the quality of our paper. We have addressed all the comments and revised our manuscript accordingly. Our responses to all the comments and the major revisions are listed below.

Point 1: Reply to Response 1. Consider explaining what this study would contribute to patient care.

Response 1: Thank you for this comment. The manuscript before revision did not explain the influence and contribution to the spastic hemiplegic cerebral palsy population. To give a better explanation of patient care, I added the paragraph below.

Before (Line 264-266 before modification)

To the best of our knowledge, this study is the first to analyze the differences in gait adaptation patterns between the legs of children with spastic hemiplegic CP according to changes in the slope angle.

After (Line in 263-265 after modification) : Deleted

After (Line in 281-289 after modification)

To the best of our knowledge, this is the first study to analyze the differences in gait adaptation patterns between the legs of children with spastic hemiplegic CP according to changes in the slope angle, which was different from slope gait adaptation mechanism of spastic diplegic CP and TD peers. Consideration of this unique adaptation pattern would be helpful to provide a specific intervention program for hemiplegic CP. For example, training focused on promoting propulsion strength of the affected leg might reduce the asymmetry of gait during incline walking in children with hemiplegic CP. However, further clinical study is needed whether these interventions can improve gait patterns on slopes.

Before (Line 284-286 before modification)

Further well-established studies with larger populations for each gait pattern classification are needed to better elucidate the mechanism during uneven-level gait.

After (Line in 293-296 after modification)

Further well-established studies with larger populations for each gait pattern classification are needed to better elucidate the mechanism during uneven-level gait and inspiration for appropriate intervention.